# Paper-Based Electrochemical Biosensors for Food Safety Analysis

**DOI:** 10.3390/bios12121088

**Published:** 2022-11-28

**Authors:** Bambang Kuswandi, Mochammad Amrun Hidayat, Eka Noviana

**Affiliations:** 1Chemo and Biosensors Group, Faculty of Farmasi, University of Jember, Jember 68121, Indonesia; 2Department of Pharmaceutical Chemistry, Faculty of Pharmacy, Universitas Gadjah Mada, Yogyakarta 55281, Indonesia

**Keywords:** electrochemical detection, paper-based device, biosensor, food safety, foodborne pathogens, rapid measurement

## Abstract

Nowadays, foodborne pathogens and other food contaminants are among the major contributors to human illnesses and even deaths worldwide. There is a growing need for improvements in food safety globally. However, it is a challenge to detect and identify these harmful analytes in a rapid, sensitive, portable, and user-friendly manner. Recently, researchers have paid attention to the development of paper-based electrochemical biosensors due to their features and promising potential for food safety analysis. The use of paper in electrochemical biosensors offers several advantages such as device miniaturization, low sample consumption, inexpensive mass production, capillary force-driven fluid flow, and capability to store reagents within the pores of the paper substrate. Various paper-based electrochemical biosensors have been developed to enable the detection of foodborne pathogens and other contaminants that pose health hazards to humans. In this review, we discussed several aspects of the biosensors including different device designs (e.g., 2D and 3D devices), fabrication techniques, and electrode modification approaches that are often optimized to generate measurable signals for sensitive detection of analytes. The utilization of different nanomaterials for the modification of electrode surface to improve the detection of analytes via enzyme-, antigen/antibody-, DNA-, aptamer-, and cell-based bioassays is also described. Next, we discussed the current applications of the sensors to detect food contaminants such as foodborne pathogens, pesticides, veterinary drug residues, allergens, and heavy metals. Most of the electrochemical paper analytical devices (e-PADs) reviewed are small and portable, and therefore are suitable for field applications. Lastly, e-PADs are an excellent platform for food safety analysis owing to their user-friendliness, low cost, sensitivity, and a high potential for customization to meet certain analytical needs

## 1. Introduction

Food safety monitoring is a very crucial factor when handling food products. Nowadays, food monitoring has become one of the major public issues globally due to the rapid changes in lifestyle, food habits, and global supply chains [1]. Since global supply chains for food are significantly increasing, food safety monitoring systems in each country are mandatory [2]. Furthermore, these systems including food quality control are necessary for consumer protection and the food industry itself, where quantification of food ingredients and contaminant screening is compulsory [3]. Some standard procedures are proposed by regulatory agencies like the EU Food Safety Authority and the US Food Drug Administration, where maximum levels for certain contaminants in food are set to protect consumers [4].

Food analysis is commonly performed with conventional techniques, e.g., chromatography, spectrometry, etc. [5,6]. These classical techniques have some drawbacks, such as being expensive, laborious, time-consuming, and requiring high sample volumes and skilled personnel [7]. Moreover, analysis using the techniques is often performed at the final stage of the food production, making it challenging to detect in which stage the contamination occurs. To address these drawbacks, electrochemical biosensors offer a low-cost approach to sample screening for potential contaminations during any stages of the food production process.

In this regard, electrochemical biosensors match several features required for in-process control such as being portable, allowing on-site measurements, minimum sample pretreatment and reagent consumption, and avoiding the use of organic solvents. Indeed, by integrating paper-based analytical devices (PADs) or paper-based devices with the electrochemical biosensor, a revolutionary step has been achieved which in turn opens up new strategies in the biosensor development by providing eco-friendly, plastic-free sensors. Furthermore, paper-based electrochemical biosensors operate by capillary force (and thus require no external pump for the fluid flow) [8], reduce reagent consumption (i.e., the reaction occurs in very small volumes (few μL) within the cellulose network) [9], are capable of storing reagents within the devices [7,10], avoid sample pretreatment by taking the advantage of the paper porosity [11], can be easily modified nanomaterials [12,13], enable multiple analyses to be performed in a simple manner [14], improve sensitivity via a multi-stage sample introduction in the paper 3D network [12], and address the drawbacks of polyester or alumina-based sensors by detecting gas analytes on the electrode surface without an external sampling system [15,16].

In the development of PADs, the Whiteside group [17,18], who also introduced PAD for the first time, is the pioneer in colorimetry-based PADs, whereas the electrochemical PAD (e-PAD) was first demonstrated by the Henry group [19]. The reported e-PAD offers high sensitivity, can be operated with miniaturized commercial electrochemical instruments, and is suitable for the analysis of colored samples, which overcomes some limitations of the colorimetric PADs. Over the last decade, the field has grown rapidly, as shown by several reviews covering recent trends in e-PAD biosensors [20,21]. However, most of these reviews generally discuss the design, fabrication, and various applications [22] of e-PADs with only little focus given to their applications in food safety [21,23]. To the best of our knowledge, this is the first review that focuses on the design, fabrication, and utilization of e-PADs for food safety analysis and monitoring.

## 2. Paper Design and Fabrication

### 2.1. Paper Types

Paper is used as a substrate in PADs because of its liquid-wicking capability, porosity, and surface affinity to various reagents, molecules, and analytes [4]. Different paper materials have different characteristics, e.g., grade, thickness, sizes, pore size, and wicking rate. The selection of paper type is necessary to fully explore the properties of the paper for increasing the analytical performance of the proposed PAD. Depending on the target analyte [18], a wide range of paper types, e.g., filter, chromatography, printer, and office paper, can be selected to construct PADs for food safety analysis.

The e-PAD biosensors commonly employ two types of paper as the substrate, i.e., high- and low-adsorption paper [24] (Table 1). The first type is represented by chromatography and filter papers, such as the Whatman filter papers. The advantage of using the high-adsorption paper is the loading capacity within the paper cellulosic structure, which in turn only needs a very low sample volume (few μL). Herein, the electrochemical cell is constituted by the paper’s porosity that allows it to avoid species diffusion at the electrode surface, which causes lower sensitivity. The second is represented by copy paper or printer paper, with low adsorption properties. This type of paper allows direct contact between the printed electrodes and the solution, providing higher sensitivity. In low adsorption paper, reagents can be introduced only onto the conductive ink, and not within the paper substrate. Both types of paper can be used as an eco-friendly substrate for biosensing platform development compared to the commonly used plastic substrates.

So far, filter paper has been the most widely used paper type. Other paper types employed in e-PADs include copy or office paper, art paper, and cellulose acetate filter paper. In addition, nitrocellulose membranes, mixed cellulose ester (MCE) membranes, and PVDF filter membranes can also be employed as substrates for PADs. The most popular paper type in the PAD fabrication is Whatman No 1 filter paper. This is due to its high alpha-cellulose content, homogeneity, smooth surface, and absence of additives, offering excellent quality, reproducibility, and uniformity for PAD fabrication [25]. Another important feature of this paper is its suitability for the immobilization of various bio-receptors, such as enzymes, antigen/antibodies, DNA, and aptamers because of its high level of non-specific binding with biomolecules [23].

**Table 1 biosensors-12-01088-t001:** Types of paper used as substrates for several paper-based electrochemical biosensors.

No.	High-Adsorption	Electrochemical Technique	References	Low-Adsorption	Electrochemical Technique	References
1	Whatman No.1 filter paper	Differential pulse voltammetry	[22]	Cellulose acetate filter paper	Cyclic voltammetry & amperometry	[26]
2	Whatman chromatography paper (3 mm)	Differential pulse voltammetry	[27]	Mixed cellulose ester (MCE)	Cyclic voltammetry	[28]
3	Whatman RC60 regenerated membrane filter	Cyclic voltammetry	[29]	Office paper	Electrochemical impedance spectroscopy	[30]
4	Filter papers (102, 15 mm)	Cyclic voltammetry and chronoamperometry	[31]	Art paper	Linear sweep voltammetry	[32]
5	Labor filter paper (67 g/m^2^)	Cyclic voltammetry & chronoamperometric	[33]	PVDF filter membrane	Cyclic voltammetry & differential pulse voltammetry	[34]
6	Nitrocellulose membrane	Cyclic voltammetry & differential pulse voltammetry	[35]	Copy paper (80 g/m^2^)	chronoamperometry	[19]

### 2.2. Device Fabrication

The e-PAD fabrication techniques, which include barrier patterning, electrode production, electrode modification utilizing nanomaterials, etc, may vary depending on the target analytes. Barrier patterning in PADs aims to discriminate the hydrophilic and hydrophobic areas, which is necessary to avoid the solution back-flow and to allow biochemical reactions to take place in the desired zone. Device fabrication, for example, can be carried out by etching or cutting. Once the paper type is selected, a defined sensing zone/area can be created by punching or cutting according to design. The size of the sensing zone and other designated areas (e.g., reaction zones) can be suited according to the sample/reagent volume. Typically, the smaller the sample volume, the smaller the sensing zone required.

Alternatively, patterning techniques such as wax printing, laser printing, inkjet printing, screen printing, flexographic printing, plasma treatment, and photolithography may be employed. The electrodes can be fabricated using various materials, e.g., carbon, graphite, gold, silver, nickel, and epoxy by screen printing, inkjet printing, or pencil drawing onto the hydrophilic area of the PAD [25]. Furthermore, to improve the analytical characteristics of the PAD, surface modification of working electrodes using nanomaterials can be performed to enhance the sensitivity and specificity of the detection [23]. Examples of utilized nanomaterials include nanocomposites bonded with silver nanoparticles, gold nanoparticles, platinum nanoparticles, palladium nanoparticles, carbon nanotubes, fullerene, graphene oxide, or iron oxide nanoparticles [36]. A previous work reporting working electrode modification using nanomaterials demonstrated an enhancement in sensitivity and specificity, which in turn produced a picogram level detection limit [10].

PAD fabrication using techniques such as wax printing, wax screen printing, inkjet printing, laser printing, and photolithography is suitable for paper with low thickness. A thinner paper needs only a small amount of wax or ink to penetrate through the paper substrate to create the hydrophobic zone. Examples of fabrication techniques for PADs that have been used to electrochemically detect analytes in food samples are presented in Table 2.

### 2.3. 2D and 3D Designs

In the early development of e-PADs, the sensors were mainly designed as a 2D device based on the horizontal/lateral flow (strip) [19], or integrated as a paper disk on a sensor platform made of other materials [47] (Figure 1). The strip design can be integrated with lateral flow assay (LFA) [48], while the folding approach [14] has led to 3D designs such as stack [49], pop-up [50], origami [14], etc., allowing multiple steps/reactions to be performed within a single paper sensor. These 3D devices can facilitate sample treatment, timing control, multistep analysis, and simultaneous detection of multiple analytes. For example, an origami design has been previously reported for multiclass pesticide detection, where multiple pads coupled with paper printed electrodes were used to detect three types of pesticide. The device was operated by folding the pads, adding distilled water, and cutting the used pads after measurement [14].

The differences between the 2D and 3D designs are the placement of electrodes in the devices. In a 2D design, the three electrode configuration is printed onto a paper substrate, generally on a hydrophobic or a circular sensing zone [19]. In a 3D design, the paper is folded to make a pop-up or an origami configuration, where the working electrode (WE) is prepared on one part, while the counter electrode (CE) and reference electrode (RE) are patterned on another part of the paper [51]. The 3D design exhibits excellent homogenous reactions in the sensing zones, where fluid can flow freely in both horizontal and vertical directions [52]. In both designs, the CE is commonly fabricated to be slightly bigger than the WE and RE, so that there is no constraint in the current flow between the WE and CE. Furthermore, the WE is generally positioned close to the RE to reduce the encompassed resistance effect between the WE and RE [19]. Some examples of 3D e-PADs are given in Figure 2.

### 2.4. Patterning Hydrophobic Barriers

To create a microfluidic channel, the cellulose substrate of the paper is typically patterned using hydrophobic barriers to define the hydrophilic zones and the fluid path of the solution [25]. The hydrophobic barrier avoids the overflowing or back-flow of solution from the device [53]. The hydrophobic specific area and hydrophilic zones would be different depending on the solution volume needed. Generally, desired detection methods may need different areas of WE, RE, and CE. Various techniques could be used for the patterning of hydrophobic barriers on a paper substrate, e.g., wax, inkjet, laser, flexographic, and screen printing, including other techniques, such as laser treatment, plasma treatment, wet etching, and photolithography [54] (Figure 3).

In the e-PADs system, the most commonly used patterning technique is wax printing, then inkjet printing, followed by laser printing, photolithography, and screen printing. Each patterning technique has advantages and disadvantages. Wax printing is the most common patterning technique because of its excellent creation of hydrophobic and hydrophilic zones on a paper substrate. It is a relatively simple approach with high flexibility in patterning hydrophobic barrier design [55]. In this technique, an office wax printer was employed to print the hydrophobic pattern onto a paper surface. The wax-printed paper was then heated in an oven (100 °C), allowing the wax to diffuse through the paper to form the hydrophobic barrier. In addition, the wax printing technique was applied to construct hydrophobic zones around hydrophilic regions at enzyme and substrate pads in 3D origami and flower-like origami biosensors [14,38]. A similar technique was applied for making hydrophobic barriers in the pop-up DNA [49,50] and the electrochemical lateral flow assay (e-LFA) devices [48].

Similarly, inkjet printing is allowed to create flexible and precise pattern designs compared to wax printing, and it is categorized as cost-effective in patterning channels on paper matrices. However, inkjet printing needs additional steps for the layering of different materials sequentially [24]. Wax printing starts printing wax onto paper substrates by a wax printer, which allows the printing of several design patterns in a minute. Then, the patterned paper is heated, making the wax melt, spreading laterally and vertically producing hydrophobic barriers through the paper thickness [56]. While screen printing, a designed screen was prepared with a pattern as a mask via which wax or ink is passed via the screen onto paper [57]. If the wax is employed, then it is melted subsequently to penetrate the paper thickness to create hydrophobic barriers. However, the wax heating process needs additional cost so that readily commercial polymers, e.g., sealant or rubber-based paint could be employed as alternatives for screen printed onto a paper surface for the producing microfluidics channels [55].

Laser printing is another popular approach that is digital printing which uses a laser beam and a toner (powdered ink). In the printing process, the beam scans back via a photoreceptive drum to create static electricity that attracts the ink. Later, the ink is fused permanently onto a paper surface. This solid ink technique is suitable for printing electronic circuits on office paper [58]. Photolithography is another alternative technique that could be used, where it uses a UV lamp exposure via a photomask of paper covered by a photoresist (light-sensitive polymer). Then a solvent is employed to remove uncured photoresist to create hydrophilic channels within hydrophobic photoresist [59]. However, the drawback of this technique is that it is expensive considering the use of photomasks and instruments. Other printing techniques, e.g., a laser cutter, could also be employed to produce hydrophobic barriers on a paper matrix [60]. The cutting could also be used free-hand as paper is cut into certain patterns without creating hydrophilic channels within hydrophobic walls [61].

### 2.5. Electrode Fabrication

A paper-based electrochemical sensor typically uses a screen-printed electrode (SPE) which is developed by screen printing conductive inks onto paper surfaces (Figure 4). The SPE generally consists of a three-electrode system. A graphite-based ink was printed on a hydrophilic zone to create the counter and working electrodes, while an Ag/AgCl ink was printed to make the pseudo-reference electrode [14,38]. Here, the cellulose fiber network of paper acts as the matrix for redox deposition and the electrode conductive materials [24]. The electrode material choices create the conductivity and sensitivity, including the device cost and the potential of applying a suitable immobilization process. The electrode materials employed for the construction of WE, CE, and RE could be different depending on the analysis purpose, the target analyte, and the samples [24]. Electrodes could be fabricated by using conducting material pastes, i.e., carbon, silver, gold, platinum, graphene, and heavy metal. The most popular used material is carbon paste in the construction of WE and CE using the screen printing technique. Carbon paste has lower interference compared to other material pastes. RE is mostly constructed by using silver paste because of its stability and constant potential for producing an excellent signal in electrochemical detection [62].

For electrode fabrication, it is better that it be prepared using the mass-production approach, e.g., screen and inkjet printings, as they allow mass production with excellent reliable printing devices [25]. Screen printing employs a designed screen to create a specific pattern of conducting paste on PADs, while inkjet printing uses the droplet deposition on a paper matrix via a computer control that needs costly equipment [63]. For instance, the SPE was constructed by screen printing graphene ink onto a PVC substrate [48], while in the pop-up DNA device, the SPE was fabricated by printing carbon graphene ink onto cellulose paper using a similar technique [50]. In the DNA device, the SPE was constructed by printing the carbon ink using the stencil printing technique [49]. Alternatively, other techniques that can be used as low-cost, simple, and equipment-free electrode constructions use a graphite pencil-drawing or a carbon ink painting on the hydrophilic zones of PADs [64]. However, these techniques are not suitable for mass-production purposes.

### 2.6. Surface Modification

After the electrodes are constructed, further electrode surface modification is needed using micro- or nanomaterials to improve PADs’ analytical performances, such as sensitivity, reproducibility, and stability. This could be achieved by surface modification, for instance, by immobilizing recognition elements for the mobile particle surfaces, causing more recognition elements captured over the static surfaces. In addition, the particle binding capacity can be strengthened and the target analyte binding interaction can be sped-up over stationary surfaces. Furthermore, signal amplification by enzymatic aid is not necessary in micro- or nano-modification on sensor surfaces [65]. In this case, a tiny amount of target analytes in the sample may often need modification with more complex nanomaterials [25].

Typical electrode surface modification approaches on e-PAD biosensors are presented in Figure 5. A direct approach for the preparation of screen-printed carbon electrode (SPCE) surface by casting the enzyme solution or graphene–enzyme composite (Figure 5a). In another approach, the enzyme was applied by drop cast over a layer of graphene, polyvinylpyrrolidone, and polyaniline (PANI) nanocomposite onto the SPCE surface (Figure 5b). This produced a droplet-like nanostructure with a high surface area that allowed a significant enhancement of biosensor sensitivity as implemented in the development of a cholesterol biosensor, based on the ChOx enzyme [66]. Another direct method for the SPCE surface modification was by casting the enzyme solution or enzyme/redox mediator mixture and drying it (Figure 5c). This approach was implemented in the development of glucose biosensors [46]. Another approach was conducted by modifying the working and counter electrodes impregnated with the redox mediator (Prussian Blue) that was mixed with the carbon ink before the SPCE was printed (Figure 5d) [19]. In another approach in the development of an ethanol biosensor, the electrodes were spotted with 3-aminopropyldimethylsiloxane to improve wettability and amplify the signal [67]. The detection mixture consisting of alcohol dehydrogenase (ADH), the cofactor, nicotinamide adenine dinucleotide (NAD+), and potassium ferricyanide as a redox mediator were then cast and dried on the electrode surface (Figure 5e).

Currently, one of the simple surface modification approaches is using nanoparticles (NPs), which can enhance the electrode surface [64]. When metal NPs are employed, they act as catalysts and charge carriers for electrochemical reactions, improving the electrode surface area and conductivity, which in turn produces high signal amplification. In the case of NPs, they could be immobilized with enzymes for further signal amplification; however, they have limited stability because of the more rapid catalytic reaction time, unless they have been modified to improve stability [65]. Furthermore, the electrode surface modification can be characterized by using some popular characterization techniques, e.g., transmission electron microscopy (TEM) for internal composition, scanning electron microscopy (SEM) for surface composition, and electrochemical impedance spectroscopy (EIS) to measure the electrochemical response toward an applied potential.

The use of gold nanoparticles (AuNPs) represents one of the commonly employed modifications for electrochemical detections on PADs. Besides the effect on increasing surface area and conductivity, they also enhance biocompatibility [68]. AuNPs could additionally be used for aptamer capture via gold-thiol chemistry for the detection of DNA and protein [69]. Moreover, AuNPs can be used in combination with materials such as graphene [70]. In another report, AuNPs were also conjugated to palladium and concanavalin-A bioconjugates to improve detection [71]. To detect acetaminophen in the presence of ascorbic acid, AuNPs conjugated with polyglutamic acid and single-wall carbon nanotubes were employed [35]. An e-PAD was developed by functionalizing the mixed cellulose ester (MCE) filter paper with graphene oxide [28]. The resulting chips were cut and linked with Cu-wires using rapid-drying Ag-paint and covered with parafilm to create the graphene/MCE working electrodes. Then, AuNPs were electro-deposited on the surface of the graphene/MCE electrodes to construct the AuNPs/graphene/MCE electrodes. The developed electrode showed greater performance in nitrite detection than that obtained by the commercial gold electrode and glassy carbon electrodes (GCEs) which is indicated by the improved voltammetric response due to the thin layer diffusion at the conductive AuNPs/graphene film electrodes, while the planar diffusion signature was observed at the bare electrodes. Aside from Au, other non-Au nanocomposites such as platinum nanoparticles (PtNPs) were implemented as signal amplification, where electron conduction was enhanced by the electrode surface modification with PtNPs that was used as matrix, with metal ions loaded on L-cysteine capped flower-like AuNPs [49]. Another approach was implemented via the deposition of an electrically conductive ink based on copper nanoparticles (CuNPs), graphite, and polystyrene [42], whereby the highest oxidation currents for glucose detection have been achieved by increasing the sensitivity of the sensor, by optimization of the ink composition with a desired experimental design.

Moreover, non-metal nanoparticle, such as graphene and carbon nanotubes, have been used in e-PAD development, in which the WEs were modified. Graphene oxide nanoflakes and zeolite nanocrystals were modified on the electrode surface of the e-PAD sensor for ketamine that show an amplified signal because of the increased surface area of the nanostructures, which in turn caused high electron transfer kinetics [37]. Another approach employed carbon black (CB) and Prussian blue nanoparticles (PBNPs) to modify the electrode of their e-PADs, which was applied to detect the H_2_O_2_ produced in the ethanol of beers [26]. The AuNP layer was modified with multi-walled carbon nanotubes (MWCNTs) for the detection of bisphenol A (BPA), where MWCNTs show significant improvement effects concerning oxidation of BPA [32].

Furthermore, the use of a novel class of 2D nanomaterials is promising in e-PAD biosensor development, such as the hybrids of MoS_2_/graphene [72], carbon dots [73], and other innovative nanomaterials that were recently developed in biosensing applications. A further improvement to increase e-PAD biosensor performance and sensitivity could be achieved by the implementation of the photonic immobilization technique, such as nanosecond and femtosecond laser pulses of antibodies onto surfaces [74] that could also be applied for other biomolecules.

## 3. Applications for Food Safety

Since they were first introduced 13 years ago, many electrochemical paper-based sensors have been developed to detect contaminants in food. These contaminants often cause foodborne illnesses such as infections and food poisoning. In addition, several proteins found in food may trigger hypersensitivity reactions in certain people. Thus, rapid analysis of food materials using electrochemical paper-based sensors could provide an additional layer of protection to prevent contaminated food from getting ingested by consumers. The following sections will discuss in more detail the applications of the sensors for detecting each class of contaminants.

### 3.1. Foodborne Pathogens

Contamination of food with foodborne pathogens presents a significant threat to public health. These pathogens are biological agents (e.g., bacteria, viruses, and parasites) that can cause a foodborne illness when ingested with food. Foodborne illness can be classified into foodborne infection (caused by ingested live pathogens that grow in the human digestive tract) and foodborne intoxication (illness caused by toxins produced by pathogens in the food products). Several common foodborne bacteria such as *Escherichia coli, Enterococcus faecalis, Listeria monocytogenes, Staphylococcus aureus*, and *Salmonella typhimurium* have been successfully detected using e-PADs (Table 3). Detection of toxins produced by *Clostridium botulinum* and norovirus-specific DNA has also been reported [75,76].

*E. coli* O157:H7 is a major pathogen often found in healthy cattle [77]. It can be transmitted to humans through bovine food products and fresh produce contaminated by bovine waste. Detection of *E. coli* O157:H7 in meat and vegetables has been demonstrated via immunoassays and enzymatic assays [78,79,80]. For example, Adkins et al. developed a simple detection method for *E. coli* based on the activity of b-galactosidase and b-glucuronidase enzymes produced by the bacteria [79]. These enzymes catalyze the conversion of their substrates into electroactive species, which can be quantified via voltammetry. Several substrates were tested and p-nitrophenyl-b-D-glucuronoside provided the lowest limit of detection (LOD) for *E. coli* determination. However, several hours of pre-enrichment were needed to attain a viable LOD (10 CFU/mL) for *E. coli* detection in water and vegetable samples. In addition, this method is not specific to *E. coli* O157:H7, and therefore will measure the total *E. coli* present in the sample regardless of the pathogenicity. This strategy has also been applied by the group for detecting *E. faecalis* and *E. faecium* based on the activity of b-glucosidase.

More specific detection of *E. coli* O157:H7 via immunoassay was reported by Wang et al. [78] using anti-*E. coli* O157:H7 antibodies immobilized on AuNP-modified graphene paper. The method can detect the pathogen down to 150 CFU/mL in ground beef and vegetable samples. A lower detection limit (4 CFU/mL) has been reported from an aptamer-based electrochemical sensor [80]. The aptamer was immobilized on a conductive paper produced by graphene paper functionalized with platinum nano cauliflower via pulsed sono electro deposition. The detection of *E. coli* O157:H7 was completed in 12 min via electrochemical impedance spectroscopy (EIS). However, only standard solutions were used in the study, and thus it would be interesting to see how different food matrices affect the analytical performance of the method. Another EIS-based aptasensor has been recently developed for detecting *L. monocytogenes* in dairy products [81]. Listeriosis, a disease caused by bacteria, can manifest as gastroenteritis, meningitis, mother-to-fetus infections, etc., and results in death in 25–30% of cases [82]. The aptasensor was established by modifying a screen-printed carbon electrode (SPCE) with tungsten sulfide (WS_2_) nanostructure, followed by aptamer immobilization. Prior to electrochemical measurement, milk and cheese samples were diluted in PBS and centrifuged to remove interferents from sample matrices.

**Table 3 biosensors-12-01088-t003:** Detection of foodborne pathogens in e-PADs.

Analyte	Detection Principle	Electrochemical Technique	Sample Matrix	Sample Volume/Size	Detection Limit	RSD	Assay Time	References
Botulinum toxin *(C. botulinum)*	Catalytic activity of toxin toward a synthetic peptide	SWV	Orange juice	100 mL	10 pM	<10%	4 h	[75]
*E. coli* O157:H7	Immunoassay	EIS	Ground beef and cucumber	10 g	150 CFU/mL	<15%	NS	[78]
	Aptamer-based assay	EIS	Standard solution	NS	4 CFU/mL	<5%	12 min	[80]
	Enzymatic assay	SWV	Alfalfa sprout	10 g	10 CFU/mL (after enrichment)	NS	8 h	[79]
*E. faecalis, E. faecium*	Enzymatic assay	SWV	Alfalfa sprout	10 g	10 CFU/mL (after enrichment)	NS	12 h	[79]
*L. monocytogenes*	Aptamer-based assay	EIS	Cheese and milk	NS	10 CFU/mL	<6%	NS	[81]
*S. aureus*	Immunoassay	DPV	Milk	NS	13 CFU/mL	<11%	~30 min	[83]
	DNA hybridization	DPV	Fruit juice	NS	0.1 nM	<5%	10 s (response time)	[84]
*S. typhimurium*	Immunoassay	Potentiometry	Apple juice	NS	5 cells/mL	<15%	<1 h	[85]
	Methylene blue-mediated detection of LAMP-amplified DNA	DPV	Drinking water and milk	10 mL	2 CFU/mL (water), 5 CFU/mL (milk)	<10%	NS, 45 min for LAMP	[86]
Norovirus	DNA hybridization	DPV	Standard solution	5 mL	100 fM	<5%	5 s (response time)	[76]

DPV: differential pulse voltammetry, EIS: electrochemical impedance spectroscopy, LAMP: loop-mediated isothermal amplification, SWV: square wave voltammetry, NS: not specified.

Besides aptasensors, nucleic acids have been extensively deployed in nucleic acid tests (NATs). NAT refers to a technique to detect a particular nucleic acid sequence from a species. Detection of *S. aureus* in fruit juice via NAT on an e-PAD has been demonstrated by Mathur et al. [84]. *S. aureus* is a major cause of infective endocarditis, prosthetic device infections, and infections on skin and soft tissues [87]. Single-stranded DNA (ssDNA) capture probe was immobilized on SPCE that had been previously modified with graphene nanodots and zeolite. The hybridization of *S. aureus* DNA to the capture probe was then measured via differential pulse voltammetry (DPV). The LOD of the sensor was 0.1 nM. An NAT-based e-PAD has also been reported for detecting DNA from Norovirus [76]. The DNA capture probe was immobilized on oxidized graphitic carbon nitride nanosheet-modified electrodes. The method could detect as low as 100 fM viral DNA. However, in both methods (*S. aureus* and Norovirus DNA detection), it is not clear how many bacteria or viruses need to be present in a sample before the sample could yield a positive detection.

The amount of genetic materials extracted from a bacterium or virus is typically very small. Since the ability to detect a low number of pathogens (<10 CFU) is critical for food analysis, amplification of extracted genetic materials are often required to achieve an acceptable LOD. Nucleic acid amplification techniques such as polymerase chain reaction, nucleic acid sequence-based amplification, and strand displacement amplification have been developed and can be used to assist pathogen detection [88]. An amplification-based-NAT using an origami paper device has been demonstrated by He et al. for *S. typhimurium* detection [86]. The device consisted of five layers on paper (Figure 6A): layers 1–2 for wicking the sample solution, layer 3 for DNA extraction, and layers 4–5 for loop-mediated isothermal amplification (LAMP) and electrochemical detection. To operate, the device was first flipped such that layers 1–3 and layers 4–5 were each stacked together. The sample was then added on layer 3 so that DNA could be extracted on the glass fiber while the remaining sample solution went to layers 1–2. Layer 3 was then flipped to the top of layers 4–5. Elution buffer was added to transfer extracted DNA from the glass fiber to layers 4–5. After that, layer 3 was flipped back to its previous position and layers 4–5 were sealed with an acetate film to prevent evaporation during LAMP. Upon completion of the LAMP reaction, amplification products were then electrochemically quantified via DPV using methylene blue. This method has successfully detected *S. typhimurium* down to 2 CFU/mL in drinking water and 5 CFU/mL in milk samples.

### 3.2. Pesticides

Pesticides play a major role in food production and are widely used to protect crops from various pests (e.g., insects, fungi, and weeds). Because they are intrinsically toxic, regulatory agencies (such as WHO, FAO, and EPA) set acceptable maximum limits of pesticide residue to protect consumers from their adverse effects [90]. Detection of pesticide residues in food matrices has been demonstrated in multiple paper sensors [91,92,93,94], with a couple of paper sensors using electrochemical techniques [95]. Although the majority of reported e-PADs have only interrogated pesticides in standard solutions and/or environmental samples, the sensors are also promising for pesticide analysis in food, by integrating suitable sample preparation techniques. Table 4 summarizes e-PADs used for pesticide determination.

Organophosphates and other organophosphorus compounds are among the most commonly used pesticides [99]. This class of pesticide is a potent inhibitor of choline esterase (ChE) enzymes. Inhibition to ChE can result in a cholinergic crisis, which often manifests in nerve and respiratory failure [100]. This inhibition activity has been exploited in many sensors aimed at pesticide detection. For example, Cioffi et al. recently developed an e-PAD for detecting paraoxon in soil and vegetable [95]. They screen-printed carbon electrodes on office paper, followed by electrode modification using Prussian Blue (electrocatalyst), carbon black (conductivity enhancer), and butyrylcholinesterase (BChE). BChE converts butyryl thiocholine into thiocholine, which is electroactive. The amount of thiocholine produced decreased as the concentration of inhibitor (i.e., paraoxon) increased, and this process was measured via chronoamperometry.

Similar strategies have been applied in other e-PADs to quantify paraoxon in standard solution, surface water, and aerosolized samples [14,24,38], with comparable detection limits (~1–3 ng/mL) and assay times (5–10 min). One of the devices took advantage of origami design to construct an e-PAD with separate layers for the enzyme, substrate, and electrodes (Figure 6B) [38]. The device consisted of two parts: (1) a folding part with two layers (each for substrate and electrodes) and (2) a single-layer sampling part with a pre-loaded enzyme, which can be inserted within the folding part. The sampling part is where the pesticide was captured. Upon the insertion of the sampling part into the folding part, all layers were then stacked together and distilled water was added to initiate the enzymatic reaction. This origami device has been applied to detect other pesticides including 2,4-dichlorophenoxy-acetic acid and glyphosate.

Besides chronoamperometry, enzymatic detection of pesticides on e-PADs has also been carried out via potentiometry and EIS [98,101]. The determination of three pesticides (avermectin, dimethoate, and phoxim) was demonstrated by Yang et al. based on time-dependent changes in acetylcholine esterase inhibition rate by the pesticides [101]. The enzyme inhibition reaction was monitored by EIS and the impedance spectra were subjected to principal component analysis and support vector machines to build a classification model that can differentiate the pesticides. More than 90% accuracy in pesticide identification was achieved by the model, which opens up the possibility of selectively detecting different pesticides using a single enzyme. However, the classification accuracy seemed to decline with increasing concentrations of pesticides. Hence, a further study investigating detection limits and dynamic range of the method would be valuable to establish its scope and applications.

### 3.3. Veterinary Drugs

While pesticides are contaminants of concern in plant-based food, food contamination with veterinary drugs is a major issue in animal-based food products. Similar to pesticides, there are maximum residue levels (MRLs) set for veterinary drugs (including active substances and their metabolites) in food [102]. To date, only a few papers have reported the development of e-PADs for veterinary drug determination, all of which target antibiotic detection in milk samples [83,103]. Majority of reported e-PADs deployed optical detection methods such as colorimetry [104,105,106,107] and fluorescence [108,109,110,111].

Many drugs are electroactive and therefore can be electrochemically detected via direct oxidation/reduction [37,112,113]. Direct oxidation has been applied to detect ciprofloxacin and sulfanilamide in milk using e-PADs [83,114]. Milk samples were diluted in deionized water or buffer and then subjected to electrochemical measurements. DPV was selected for the electrochemical measurements due to its high sensitivity. Similar detection limits were reported for both drugs (~5 mM). A lower detection limit (0.04 ng/mL or ~0.1 nM) has been reported for the determination of neomycin via immunoassay [115]. Antibodies to neomycin were immobilized on single-walled carbon nanotubes-modified filter paper. The binding of the target drug to its antibodies was then measured by chronoamperometry. A more rigorous sample pre-treatment was also applied in this work, where milk samples were treated with acetic acid, followed by dilution with water, shaking for 1 h, centrifugation to remove proteins, and filtration, before electrochemical measurements.

### 3.4. Allergens

Food allergies can cause various clinical complications within the gastrointestinal tract, respiratory system, and skin [116]. This abnormal immune reaction to allergens has increasingly become a major health concern, especially in developed countries [117]. Currently, eight foods are identified as major allergens by Public Law in the United States [118]. They are milk, eggs, fish, crustacean shellfish, tree nuts, peanuts, wheat, and soybeans.

An e-PAD has been developed to detect Ara h1, a peanut allergen [41]. The allergen was detected by utilizing aptamer-decorated black phosphorus nanosheets via DPV. The method has been applied to analyze Ara h1-spiked cookie dough and can detect down to 21.6 ng/mL allergen within 20 min. Another peanut allergen, Ara h2, has also been successfully detected using a whole-cell-based e-PAD developed by Jiang et al. [119]. Rat basophilic leukemia mast cells, as the sensing element, were immobilized on paper with the help of a biocompatible polymeric composite from methacryloyl-modifed gelatin (GelMA), polyvinyl alcohol (PVA) and nano-hydroxyapatite (nHAP). The presence of Ara h2 can trigger cellular degranulation and inflammatory factor release by the cells, which affect the capacitance of the sensor. This change in capacitance was then exploited to quantify the allergen molecules. Using this method, the group was able to achieve a detection limit of 0.028 ng/mL. They had also previously applied a similar approach to quantify casein, a milk allergen [39].

### 3.5. Heavy Metals

Contamination of crops with heavy metals is another major problem in assuring food safety. Heavy metals can get transferred to crops from atmospheric deposition, metal-containing pesticides, livestock manure, and irrigation with polluted water [120]. A list of e-PADs for detecting heavy metals and their performance characteristics is shown in Table 5. Many of these sensors applied direct detection where metals/metal ions were oxidized/reduced on the electrode to generate measurable current. Stripping voltammetry is often used to preconcentrate the metals on the electrode surface to achieve ng/mL (ppb) or sub ppb level of detection limit [89,121] because the maximum permissible concentrations of heavy metals in food or drinking water set by the regulation agencies are typically in low ppb levels. For example, FDA and EPA set a maximum limit of 5 ppb and 15 ppb of lead (Pb), respectively, for drinking water [122].

Electrochemical detection of Pb has been demonstrated in multiple paper sensors. Simultaneous detection of Pb and other metals such as cadmium (Cd), zinc (Zn), and/or tin (Sn) in samples is also possible due to the difference in their redox potentials [89,121]. For instance, Soulis et al. recently created an e-PAD fully drawn using an x-y plotter to simultaneously detect Pb and Cd in fish food [123]. Bismuth (Bi) was added to the sample solution to allow co-deposition of Bi and target metals since the formation of Bi-target metal alloy has been shown to improve detection sensitivity. The alloys were then deposited on the electrode by applying −2.5 V for 4 min, followed by anodic stripping. Oxidation peaks for Pb and Cd showed up at around −1.0 and −1.2 V, respectively. Prior to electrochemical measurements, samples were digested using HNO_3_-HCl (3:1 *v*/*v*) to release metal ions from organic matter. Stripping voltammetry for simultaneous detection of Pb and Cd on paper sensors has also been reported by other researchers for determination in drinking water and beverages [124,125]. For this application, samples can be directly measured on the sensors without any pre-treatment.

A sub-ppb detection limit was reported by Pungjunun et al. for the simultaneous detection of Pb and Sn in canned food [89]. They used a mixture of graphene ink and Bi NPs (2:5 *w/w*) to produce SPCE that could be used with a commercial portable potentiostat (Figure 1C). Milder sample preparation was applied where the liquid phase of the sample was diluted with 0.1 M oxalic acid and 0.1 mM cetyl trimethyl ammonium bromide (CTAB), while the solid phase was ground, mixed with 2% HNO_3_ for 5 min, and adjusted to pH 7 before dilution with oxalic acid and CTAB. The sample solution was then transferred onto the e-PAD and subjected to square-wave anodic stripping voltammetry.

Besides direct detection using stripping voltammetry, the nucleic acid-based assay has also been implemented to achieve very low detection limits on e-PADs. Qian et al. recently demonstrated an aptamer-based e-PAD that enables the detection of Cd and Pb down to pM (ppt) level in vegetable and fruit samples [126]. A capture ssDNA was immobilized on AuNP-modified electrodes, followed by the addition of complementary aptamers that had been labeled with redox reporters (i.e., ferrocene for Cd aptamer and methylene blue for Pb aptamer). Pb and Cd in the sample solution disrupted the capture ssDNA–aptamer hybrid and therefore reduced the number of redox reporters on the electrode surface. This reduction was then measured by DPV to quantify metal concentrations. The electrochemical measurement could be completed within 15 min. However, the sample preparation method and time required for the preparation were not specified in detail.

**Table 5 biosensors-12-01088-t005:** Detection of heavy metals in e-PADs.

Analyte	Detection Principle	Electrochemical Technique	Sample Matrix	Sample Volume/Size	Detection Limit	RSD	Assay Time	References
Cd(II)	Direct detection	DPV	Rice	0.2 g	0.1 ng/mL	20–40%	~1 h	[127]
Cd(II), Pb(II)	Direct detection	SWASV	Soda water	100 mL	2.3 ng/mL (Cd) 2.0 ng/mL (Pb)	<5%	4 min	[125]
		ASV	Drinking water	500 mL	2.33 ng/mL (Cd) 0.97 ng/mL (Pb)	5–10%	~20 min	[124]
		DPASV	Fish food	1 g	3.1 ng/mL (Cd) 4.5 ng/mL (Pb)	<15%	~5 min *	[123]
		DPASV/SWASV	Tap water	160 mL	2.4 ng/mL (Cd) 4.2 ng/mL (Pb)	<15%	~8 min	[128]
	Aptamer-based assay	SWV	Vegetable and fruit	60 mL	23.3 pM (Cd) 46.2 pM (Pb)	<10%	15 min *	[123]
Cd(II), Pb(II), Zn(II)	Direct detection	SWASV	River water	100 mL	1.3 ng/mL (Cd) 0.9 ng/mL (Pb) 10.5 ng/mL (Zn)	<15%	~5 min	[121]
Hg(II)	Direct detection	ASV	River water	40 mL	30 nM	<10%	~10 min	[129]
Ni(II)	Direct detection	AdCSV	Water	20 mL	6.27 ng/mL	<5%	~ 3 min	[130]
Pb(II), Sn(II)	Direct detection	SWASV	Canned food	500 mL/1 g	0.26 ng/mL (Pb) 0.44 ng/mL (Sn)	<5%	2 min *	[89]
Zn(II)	DNAzyme-based assay	DPV	Tap water	5 mL	0.03 nM	<15%	~40 min	[131]

* does not include time for sample preparation. AdCSV = Adsorptive cathodic stripping voltammetry, ASV = Anodic stripping voltammetry, DPV = differential pulse voltammetry, DPASV = differential pulse voltammetry, SWASV = square wave anodic square voltammetry, SWV = square wave voltammetry.

## 4. Conclusions, Challenges, and Prospects

Food safety analysis has become an important issue, especially with the increasing incidence of food contamination, including adulteration. e-PAD biosensors hold a great promise for developing highly sensitive and selective detection approaches needed for food safety analysis. The sensors can provide a platform for quantitative analysis, where sample preparation and treatment can be conducted on a single piece of paper. Along with these benefits, paper is well known as a cheap, readily available, and sustainable material which hopefully encourages the development of fully integrated e-PADs. Paper allows for the incorporation of nanomaterials and biomaterials and sensors made of paper can be easily fabricated using currently available manufacturing techniques. To improve the sensitivity and selectivity of e-PADs for detecting food contaminants, several approaches can be performed such as: (1) engineering the flow control and optimizing the reaction step and timing using a 2D or 3D device and (2) using suitable materials for the electrode systems such as micro- or nanomaterial to improve conductivity and electron transfer on the electrode, and (3) modifying the electrode surface with various micro- and nanomaterials to improve conductivity and increase the surface area of the electrode.

This review summarizes various developments of e-PAD biosensors for the detection of food contaminants including pathogens, pesticides, veterinary drugs, allergens, and heavy metal contaminants. The electrochemical method seems to be significantly used over other detection methods, because of its features, e.g., miniaturization, simple measurement, efficient analysis, high sensitivity and selectivity, and suitability for food safety analysis. The application of user-friendly and disposable e-PAD biosensors in the food safety analysis could offer an innovative and transformative approach for evaluating food quality during the manufacturing processes in the industry and distribution and when finally reaching consumers. New approaches and strategies are developed continuously to further improve sensitivity and selectivity toward the target analyte. Integration of nanomaterials, bioreceptor, processing methods, and detection approaches would support the wide use of the paper sensor technology.

A big challenge for avoiding the use of expensive instrumentation such as bench-top potentiostats for reading the electrochemical signal still exists. In this case, a smart phone connected to a dedicated board that serves as the source of data transmission for e-PAD analysis can be used as an alternative. Moreover, further work should be directed towards the development of instrument-free techniques and non-equipment-based readouts, which may be achieved with the aid of IoT (Internet of Things) and artificial intelligence. For example, portable and open-source potentiostats could address this issue, including the use of data transmission devices such as LE Bluetooth module for IoT applications [132]. In addition, several open-source and low-cost potentiostats, such as the UWED [133], DStat [134], and CheapStat [135], etc., have been developed that could be promising for this purpose. Lastly, improving stability, sensitivity, and storage life of e-PADs could push for future commercialization for early detection of food contaminations, where foodborne diseases are increased globally.

## Figures and Tables

**Figure 1 biosensors-12-01088-f001:**
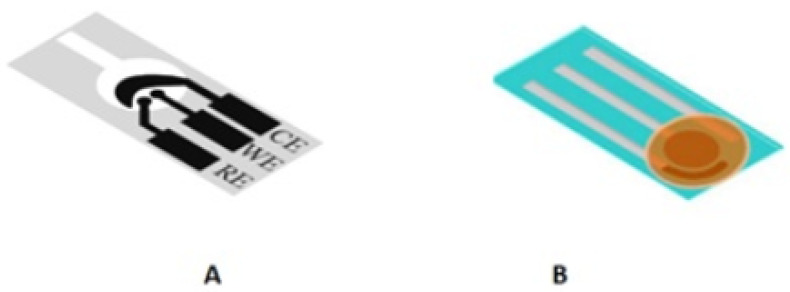
Design of 2D e-PAD biosensor. The 2D designs were adapted from the references: (**A**) strip [19], and (**B**) paper disk [47]. Counter electrode (CE), working electrode (WE), and reference electrode (RE).

**Figure 2 biosensors-12-01088-f002:**
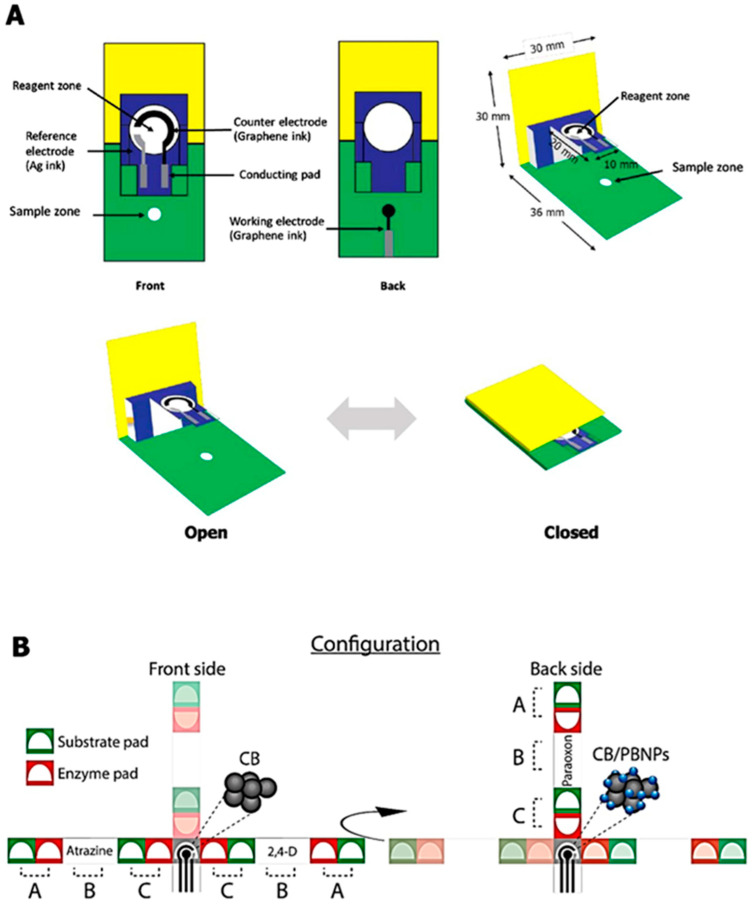
Examples of e-PADs with 3D design, (**A**) the pop-up DNA biosensor design and its operation, adapted from [50], and (**B**) the 3D-origami enzyme biosensors with their measurement steps [14].

**Figure 3 biosensors-12-01088-f003:**
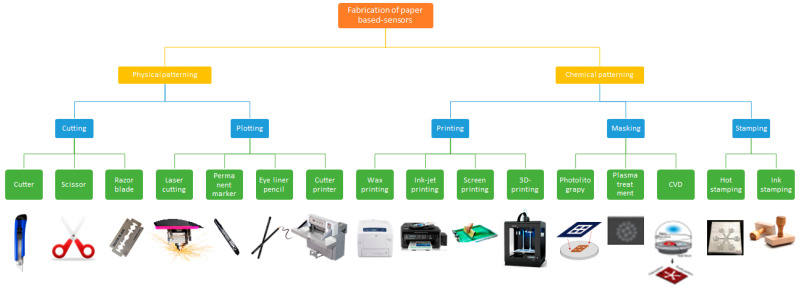
Patterning techniques that can be used for e-PAD biosensors, adopted from reference [54]. Open Access under CC BY-NC.

**Figure 4 biosensors-12-01088-f004:**
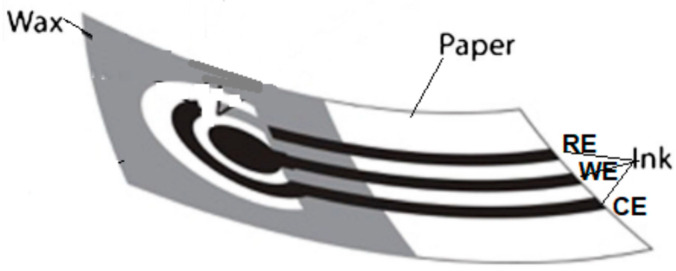
Typical screen-printed electrode (SPE) of e-PADs obtained with the screen-printed technique.

**Figure 5 biosensors-12-01088-f005:**
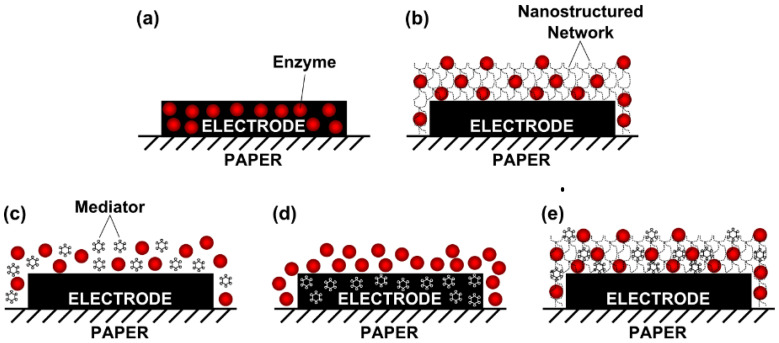
Electrode surface modification approaches of e-PAD biosensors: (**a**) enzyme-modified electrode; (**b**) nanostructured layer-modified electrode with biomolecule; (**c**) electrode modified with redox mediator and biomolecule; (**d**) electrode modified with redox mediator composite and biomolecule; (**e**) electrode modified with biomolecule and redox mediator mixture over nanostructured layer, adopted from reference [25]. Open Access under CC BY-NC.

**Figure 6 biosensors-12-01088-f006:**
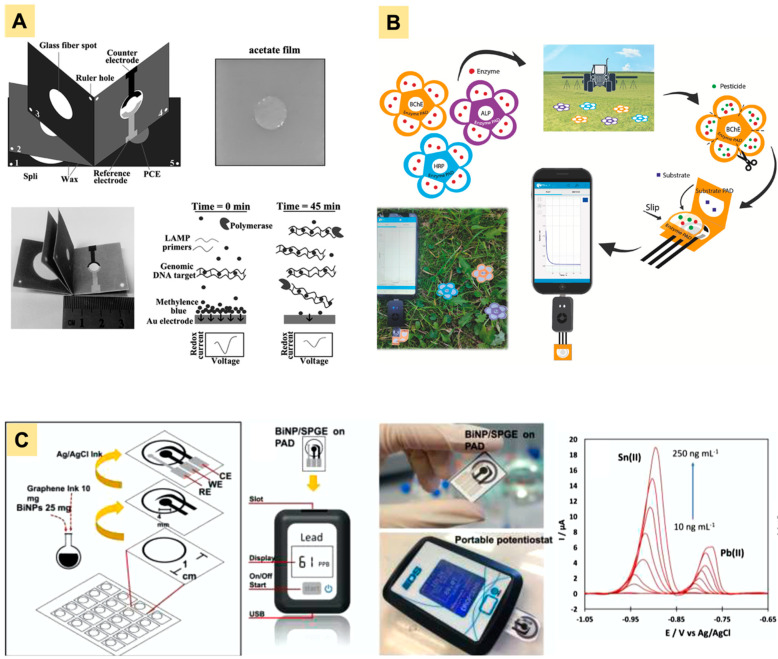
Detection of several food contaminants using e-PADs: (**A**) An origami device for *S. typhimurium* detection, adapted from ref [86]. Open Access under CC BY-NC. (**B**) A flower-like origami device for pesticide detection adapted with permission from ref [38] Copyright 2022 Elsevier. (**C**) A paper sensor for simultaneous Sn and Pb detection using a portable potentiostat adapted with permission from ref. [89]. Copyright 2022 Elsevier.

**Table 2 biosensors-12-01088-t002:** Some examples of fabrication techniques for electrochemical PAD construction.

No.	Fabrication Technique	Analyte	Sample	References
1	Wax printing	Ethanol	Beer	[26]
2	Wax printing	Ketamine	Alcoholic and non-alcoholic beer	[37]
3	Wax printing	Pesticides (insecticides and herbicides)	River water	[14]
4	Wax printing	Pesticides	Aerosol	[38]
5	Wax printing	Casein allergen	Milk	[39]
6	Wax printing	Glycoproteins	Eggs white	[40]
7	Wax Printing	Peanut allergen Ara h1	Cookie dough	[41]
8	Wax Printing	Glucose and total carbohydrate	Food stuff	[42]
8	Screen Printing	Ferricyanide	Standard solution	[43]
9	Inkjet Printing	Ascorbic acid	Dietary supplement	[44]
10	Laser Printing	Glucose	Blood	[45]
11	Photolithography	Heavy-metal ions and glucose	Aqueous solutions	[46]

**Table 4 biosensors-12-01088-t004:** Detection of pesticide residues in e-PADs.

Analyte	Detection Principle	Electrochemical Technique	Sample Matrix	Sample Volume/Size	Detection Limit	RSD	Assay Time	References
2,4-dichlorophenoxy-acetic acid	Enzymatic assay	Chrono-amperometry	River water	5 mL	50 ng/mL	<5%	~10 min	[14]
			Standard solution	NS	30 ng/mL	6%	<10 min	[38]
Avermectin, dimethoate, and phoxim	Enzymatic assay, combined with multivariate analysis	Electrochemical impedance spectroscopy	Vegetable	30 mL	NS (tested conc.: 0.1–0.3 mg/kg)	NS	15 min	[96]
Glyphosate	Enzymatic assay	Chrono-amperometry	Standard solution	NS	10 ng/mL	7%	<10 min	[38]
Malathion	Mitochondria-based assay	Cyclic voltammetry	Standard solution	NS	20 nM	~20%	NS	[97]
Paraoxon	Enzymatic assay	Chrono-amperometry	Soil and vegetable	1 g	1.3 ng/mL	<15%	<1 h	[95]
			River water	5 mL	2 ng/mL	<5%	~10 min	[14]
			Standard solution	NS	2 ng/mL	3%	<10 min	[38]
			River water and wastewater	5 mL	3 ng/mL	<15%	~5 min	[26]
Parathion	Enzymatic assay	Potentiometry	Standard solution	10 mL	0.06 nM	<10%	~10 min	[98]
Triazine	Enzymatic assay	Chrono-amperometry	River water	5 mL	NS	<5%	~10 min	[14]

NS: not specified.

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
