# Peer review of "Paper-Based Electrochemical Biosensors for Food Safety Analysis"

_biosensors, 2022, doi:10.3390/bios12121088_

Round 1
Reviewer 1 Report
Please refer to the attached file for detailed comments.

Author Response
Review 1.
This is a very timely review of paper-based biosensors with respect to food safety. The authors provided board views on paper-based device fabrication and their application on foodborne toxins and pathogens detection. The manuscript is well structured. However, the grammatical errors are so common that the clarity and credibility of the manuscript are greatly undermined. I would suggest the authors correct the major and mirror comments to make the manuscript suitable for publication.
Response: Many thanks for the positive response and suggestions. We agree and corrected the grammatical error throughout the manuscript.
Major Comments:
1. Please check and correct the many grammatical errors in the abstract, section 1 (introduction), section 2 (Paper design and fabrication), and section 4 (conclusions, challenges, and prospects). These errors made the manuscript hard to read and confusing. To name a few. page 1, line 13-14. “it is a challenge to detect ... in a rapid sensitive, portable, and user-friendly” should be “it is a challenge to detect ... in a rapid sensitive, portable, and user-friendly manner.”
Response: Many thanks for the comments and corrections. We revised the sentence accordingly.
Page 1, lines 14-15. “Recently, fast developments in paper-based electrochemical biosensors have paid attention to researchers...” should be “Recently, researchers have paid attention to the fast developments in paper-based electrochemical biosensors...”
Response: Many thanks for the comments and corrections. We revised the sentence accordingly.
Page 1, lines 18-19. “..., and hinder the sample treatment by employing the paper porosity [11].” The sentence is confusing about how the porosity could hinder sample treatment. Upon closer inspection of the cited paper. It occurs to be the immobilization of the substrate onto the paper fiber hinders the classic diffusion model of the reactance. I would recommend rephrasing or removing this phrase as the word “hinder” offers a negative meaning while the author was talking about the advantages of PADs.
Response: Many thanks for the comments and corrections. We revised the sentence by replacing hinder with avoid, accordingly.
Page 1, line 26-27. “...handling threats to human and public health.” The “human” is redundant if “public health” is the concern here. However, if the author intended to express other meanings such as “human rights”, then the phrase should be modified accordingly.
Page 1, line 37-38. “...as changes in lifestyle, food habits and the food global supply chains [1].” Maybe consider using “due to the rapid changes in lifestyle, food habits and the food global supply chains [1].”
Response: Many thanks for the comments and corrections. We revised the sentence as suggested accordingly.
Minor Comments:
1. Please check where the abbreviations were first introduced. For example, the “gold nanoparticle (AuNP)” was introduced twice, once on page 9, 329, and the other on page 12, 404-405. The latter one should be removed and just use “AuNP”.
Response: Many thanks for the comments and corrections. We revised the sentence as suggested accordingly.
Page 4, 152- 153. Please double-check the rationale that the amount of wax used in defining the hydrophobic/hydrophilic region would affect the device cost. Indeed, the wax used in a wax printer is much more expensive than candle wax. But on the device level, the cost increase from a single-layer to double-layer wax is usually much less than that of other reagents, e.g. PEDOT:PSS, AuNP etc.
Response: Many thanks for the comments and corrections. We revised the sentence accordingly.
Page 20, line 652-654. More details/citations could be helpful to clarify the kind of devices that connects to cell phones. Since the author was talking about lab bench-top potentiostats would not fit the low-cost application settings. The author could specify what kind of devices could benefit future diagnosis. For example, would portable and open-source potentiostats be helpful, or some other data transmission devices like LE Bluetooth module for IoT applications? In recent years, there were some development in open-source and low-cost potentiostats, such as the UWED from the Whitesides group, DStat from the Wheeler group, and CheapStat from UCSB, to name a few.
Response: Many thanks for the comments and corrections. We revised the paragraph accordingly.
The page number in the top right margin is not continuous. It seems to resets every time a table is inserted.
Response: Many thanks for the comments and corrections. We revised the page number accordingly.

Reviewer 2 Report
This manuscript reviews a number of research articles where paper-based electrochemical biosensors are used to monitor different categories of analytes for the Food Safety and Food Industry.
There is a good discussion on different steps to design and fabricate such paper-based biosensors using different techniques and materials; advantages and limitations are also covered. The manuscript well describes and compares the techniques.
Applications for Food Safety is discussed; Foodborne Pathogens, Pesticides, Veterinary drugs, Allergens, and Heavy Metals are the type of analytes that are considered. Detection of various analytes via enzymes, antigen, antibody, DNA, aptamer, and cells as bioreceptors using several electrochemical transduction techniques are detailed.
The use of different nanomaterials is stated as well to improve the signal. However, authors could consider to include more novel class of 2D nanomaterials that are in fact used in the biosensing application; this is important for an up-to-date review. Examples of such innovative materials used in sensing and bio-related fields can be found in RSC Adv., 7, 50166-50175 (2017) and ACS Appl. Nano Mater., 1, 1, 2–25 (2018).
A further improvement could be adding a discussion on the application of the photonic immobilization technique, also used in: Applied Physics A volume 117, pages 185–190 (2014) to improve sensitivity of the sensors; this actually could increase performance and sensitivity of such biosensors.
The manuscript is organized, and well-written. The topic is interesting and sensing in the food industry have very many interests for the researchers, and at the same times in is enormously vital and impacting in the industrial applications.
The quality of the pictures must be improved; figure 1 and figure 2, for instance, have very poor qualities.
In conclusion, given the interest and potential applications, and the quality of the presented manuscript, I support its publication in the journal of Biosensors.
Author Response
This manuscript reviews a number of research articles where paper-based electrochemical biosensors are used to monitor different categories of analytes for the Food Safety and Food Industry.
There is a good discussion on different steps to design and fabricate such paper-based biosensors using different techniques and materials; advantages and limitations are also covered. The manuscript well describes and compares the techniques.
Applications for Food Safety is discussed; Foodborne Pathogens, Pesticides, Veterinary drugs, Allergens, and Heavy Metals are the type of analytes that are considered. Detection of various analytes via enzymes, antigen, antibody, DNA, aptamer, and cells as bioreceptors using several electrochemical transduction techniques are detailed.
Response: Many thanks for the encouraging response, we really appreciate it.
The use of different nanomaterials is stated as well to improve the signal. However, authors could consider to include more novel class of 2D nanomaterials that are in fact used in the biosensing application; this is important for an up-to-date review. Examples of such innovative materials used in sensing and bio-related fields can be found in RSC Adv., 7, 50166-50175 (2017) and ACS Appl. Nano Mater., 1, 1, 2–25 (2018).
Response: Many thanks for the comments and suggestions. We agree and revised this part as suggested accordingly.
A further improvement could be adding a discussion on the application of the photonic immobilization technique, also used in: Applied Physics A volume 117, pages 185–190 (2014) to improve sensitivity of the sensors; this actually could increase performance and sensitivity of such biosensors.
Response: Many thanks for the comments and suggestions. We agree and revised this part as suggested accordingly.
The manuscript is organized, and well-written. The topic is interesting and sensing in the food industry have very many interests for the researchers, and at the same times in is enormously vital and impacting in the industrial applications.
Response: Many thanks for the supportive response, we really appreciate it.
The quality of the pictures must be improved; figure 1 and figure 2, for instance, have very poor qualities.
Response: Many thanks for the comments and suggestions. We agree and revised these figures as suggested accordingly.
In conclusion, given the interest and potential applications, and the quality of the presented manuscript, I support its publication in the journal of Biosensors.
Response: Many thanks for the supportive response, we really appreciate it.

Round 2
Reviewer 1 Report
Thanks to the hard-working authors for modifying the paper in such a short time. I think this review paper is well-written and well-suited for publication. This paper will be a great source of reference on paper-based detectors of toxins and pathogens.
Best wishes to the authors on their academic endeavors.